# Validation of Liquid Chromatography-Tandem Mass Spectrometry-Based 5-Plex Assay for Mucopolysaccharidoses

**DOI:** 10.3390/ijms21062025

**Published:** 2020-03-16

**Authors:** Tsubasa Oguni, Shunji Tomatsu, Misa Tanaka, Kenji Orii, Toshiyuki Fukao, Jun Watanabe, Seiji Fukuda, Yoshitomo Notsu, Dung Chi Vu, Thi Bich Ngoc Can, Atsushi Nagai, Seiji Yamaguchi, Takeshi Taketani, Michael H. Gelb, Hironori Kobayashi

**Affiliations:** 1Clinical Laboratory Division, Shimane University Hospital, Izumo 693-8501, Japan; toguni@med.shimane-u.ac.jp (T.O.); ynotu34@med.shimane-u.ac.jp (Y.N.); 2Department of Pediatrics, Shimane University Faculty of Medicine, Izumo 693-8501, Japan; Shunji.Tomatsu@nemours.org (S.T.); sfukuda@med.shimane-u.ac.jp (S.F.); seijiyam@med.shimane-u.ac.jp (S.Y.); ttaketani@med.shimane-u.ac.jp (T.T.); 3Nemours/Alfred I. DuPont Children’s Hospital, Wilmington, DE 19803, USA; 4Shimadzu Corporation, Kyoto 604-8442, Japan; misa.tanaka@hotmail.co.jp (M.T.); jun_wtnb@shimadzu.co.jp (J.W.); 5Department of Pediatrics, Graduate School of Medicine, Gifu University, Gifu 501-1193, Japan; kenjior-gif@umin.ac.jp (K.O.); toshi-gif@umin.net (T.F.); 6Department of Medical Genetics and Metabolism; Center for rare disease and Newborn Screening, National Children’s Hospital, Hanoi 18/879, Vietnam; dungvu@nch.org.vn (D.C.V.); bscanbichngoc@gmail.com (T.B.N.C.); 7Department of Neurology, Shimane University Faculty of Medicine, Izumo 693-8501, Japan; anagai@med.shimane-u.ac.jp; 8Departments of Chemistry and Biochemistry, University of Washington, Seattle, WD 98195, USA; gelb@chem.washington.edu

**Keywords:** mucopolysaccharidosis, newborn screening, enzyme assay, liquid chromatography tandem mass spectrometry

## Abstract

Mucopolysaccharidoses (MPSs) are rare lysosomal storage diseases caused by the accumulation of undegraded glycosaminoglycans in cells and tissues. The effectiveness of early intervention for MPS has been reported. Multiple-assay formats using tandem mass spectrometry have been developed. Here, we developed a method for simultaneous preparation and better measurement of the activities of five enzymes involved in MPSs, i.e., MPS I, MPS II, MPS IIIB, MPS IVA, and MPS VI, which were validated using 672 dried blood spot samples obtained from healthy newborns and 23 patients with MPS. The mean values of the enzyme activities and standard deviations in controls were as follows: α-iduronidase (IDUA), 4.19 ± 1.53 µM/h; iduronate-2-sulfatase (I2S), 8.39 ± 2.82 µM/h; *N*-acetyl-α-glucosaminidase (NAGLU), 1.96 ± 0.57 µM/h; *N*-acetylgalactosamine-6-sulfatase (GALNS), 0.50 ± 0.20 µM/h; and *N*-acetylgalactosamine-4-sulfatase (ARSB), 2.64 ± 1.01 µM/h. All patients displayed absent or low enzyme activity. In MPS I, IIIB, and VI, each patient group was clearly separated from controls, whereas there was some overlap between the control and patient groups in MPS II and IVA, suggesting the occurrence of pseudo-deficiencies. Thus, we established a multiplex assay for newborn screening using liquid chromatography tandem mass spectrometry, allowing simultaneous pretreatment and measurement of five enzymes relevant to MPSs.

## 1. Introduction

Mucopolysaccharidoses (MPSs) are a group of lysosomal storage diseases (LSDs) caused by a deficiency of lysosomal enzymes, leading to the accumulation of undegraded glycosaminoglycans (GAGs) such as chondroitin sulfate, dermatan sulfate, heparan sulfate, and keratan sulfate in the cells and tissues. GAGs accumulate in multiple organs, such as the visceral organs, respiratory organs, cardiovascular tissues, and osteocartilaginous tissues, and in the central nervous system. MPSs are classified into seven types which are caused by deficiencies in 11 different enzymes. The frequency of each disease type differs by race. Although MPS I (MIM 607014, 607015, 607016) and III (MIM 252900, 252920, 252930, 252940) are common in Europe and the United States, MPS II (MIM 309900) is most common in most East and Southeast Asian countries, including Japan. The frequency of all MPS types in Japan has been estimated as 1 in 63,000, of which 53% are MPS II and 13% are MPS I [1,2]. Symptoms of patients with MPS are generally severe and progressive. Most patients develop symptoms within 3 years after birth but typically lack symptoms during newborn periods. However, it was reported that GAGs accumulate in patients with MPS I, II, III, IVA (MIM 253000), and VI (MIM 253200) as early as the embryonic stage [3,4,5,6]. Increased levels of GAGs have also been reported in dried blood spots (DBS) in the neonatal period [7,8,9,10]. The effectiveness of enzyme replacement therapy and hematopoietic stem cell transplantation for patients with MPS following early detection has also been reported [11,12,13,14,15,16,17]. Early treatment results in a better quality of life; thus, early detection is important. Newborn screening (NBS) for MPS I is performed in several countries [18].

NBS methods for MPS include measurement of enzyme activities and/or accumulated GAGs in DBS [7,10,18,19,20,21,22,23,24,25,26,27,28,29]. When measuring the enzymatic activities for MPS II, IV, and VI, the in-source fragmentation will affect the quantification of enzyme activity. In-source fragmentation is a phenomenon of generating unwanted byproducts of identical mass to reaction product from substrate at heating nebulizer. Therefore, column separation is necessary to analyze enzyme activities for these MPS types. Liu et al. reported a multiplex liquid chromatography tandem mass spectrometry (LC-MS/MS) assay for simultaneous measurement of the enzyme activities of MPS I, II, IIIB, IVA, VI, and VII [30]. However, the report did not include the results of analysis with actual samples for MPS I. Similarly, current screening of MPS I and other MPSs, including MPS II, is performed using separate assays in many regions. The reason can be considered to be that MPS I screening can be analyzed by flow injection as well as several LSDs, whereas screening for MPS II, MPS IVA, and MPS VI requires column separation by LC. However, in Asian countries, including Japan, the frequencies of MPS II and MPS I are high. Performing multiple assays in a single well simultaneously is desirable in terms of cost and throughput. Accordingly, in this study, we developed an analytical method in our laboratory for simultaneous pretreatment and better quantification of the activities of five enzymes important in MPS—α-iduronidase (IDUA, EC 3.2.1.76) in MPS I, iduronate-2-sulfatase (I2S, EC 3.1.6.13) in MPS II, *N*-acetyl-α-glucosaminidase (NAGLU, EC 3.2.1.50) in MPS IIIB, *N*-acetylgalactosamine-6-sulfatase (GALNS, EC 3.1.6.4) in MPS IVA, and *N*-acetylgalactosamine-4-sulfatase (ARSB, EC 3.1.6.12) in MPS VI—by using LC-MS/MS. We validated the practicability of the methods using DBS samples obtained from healthy newborns and patients with MPS.

## 2. Results

### 2.1. Chromatograms for Each Disease Type

A typical LC-MS/MS chromatogram is shown in Figure 1. The chromatograms of each substrate, internal standard (IS), and reaction product showed good peak shapes and were clearly separated. For each enzyme, the product and chemically identical, but isotopically substituted, IS co-eluted from the LC column. In each case, the substrate was eluted from the LC column at a different retention time than the product/IS. Thus, any conversion of substrate to product in the heated electrospray ionization source of the mass spectrometer was of no consequence because only the peak representing the enzymatically produced product was integrated.

### 2.2. Enzyme Activities in Controls and Patients with MPS

Scatter plots of the enzyme activities in 672 control samples and patients with MPS are shown in Figure 2. The activities (mean ± standard deviation, 99.5 percentile) of IDUA, I2S, NAGLU, GALNS, and ARSB in control samples were as follows: IDUA (4.19 ± 1.53 µM/h, 1.77 µM/h), I2S (8.39 ± 2.82 µM/h, 1.08 µM/h), NAGLU (1.96 ± 0.57 µM/h, 0.95 µM/h), GALNS (0.50 ± 0.20 µM/h, 0.17 µM/h), and ARSB (2.64 ± 1.01 µM/h, 0.95 µM/h), respectively.

In the histograms for all types of MPS enzymes, the shapes of the histograms indicated a highly skewed distribution. However, in the histogram for MPS II, there were some control samples that exhibited very low enzymatic activity that were well separated from the bulk of the other samples (Figure 2). The activities in patients with different MPS types were as follows: IDUA in MPS I (mean: 0.428 µM/h; value: 0.414–0.442, *n* = 2), I2S in MPS II (mean: 0.085 µM/h; range: 0.051–0.203, *n* = 14), NAGLU in MPS IIIB (0.046 µM/h, *n* = 1), GALNS in MPS IVA (mean: 0.031 µM/h; range: 0–0.125, *n* = 4), and ARSB in MPS VI (mean: 0.232 µM/h; value: 0.230–0.234, *n* = 2), respectively. The enzyme activity of each patient group with MPS I, IIIB, and VI was clearly separated from that of the control group. However, there were some overlaps between the control and patient groups for MPS II and IVA. Each patient group for MPS II and IVA was compared to controls using t-tests. Significant differences were observed between control and patient groups, with *p* < 0.0001 for MPS II and *p* < 0.0001 for MPS IVA.

### 2.3. Analysis of Stability

The mean enzyme activities in quality control (QC) and healthy controls analyzed on five different days are shown in Table 1. Each QC sample was measured four times in each batch. Most values of the coefficient of variation (CV) in the intraday assay were less than 15%; however, the CV of the interday assay fluctuated from 6.1% to 24.7%, with a higher proportion of more than 15% observed for IDUA, GALNS, and ARSB (Table 2).

## 3. Discussion

In this study, we established an enzyme activity assay using LC-MS/MS for simultaneous pretreatment and measurement of five types of enzymes lacking in MPS I, MPS II, MPS IIIB, MPS IVA, and MPS VI. This work is an extension of a previous study by Liu et al. [30]. In the present study, we developed the analytical method of better LC separation by changing the analytical time from 2 min to 6 min to improve measurement sensitivity of enzyme activities for five MPSs including MPS I and II. The samples were pretreated in the same well and analyzed simultaneously when analyzing enzyme activities for five MPSs.

NBS for MPS has mostly focused on MPS I. The flow-injection assay has been used to screen for MPS I along with other LSDs. However, the flow-injection method is not appropriate for screening MPS II, MPS IVA, and MPS VI because their substrates limit the ability to quantify the enzyme reaction products [32]. Therefore, separating the product from the substrate is necessary. Our method overcomes this limitation by using column separation, in which the product peak eluted earlier than the substrate. Our method allows for simultaneous preparation and analysis of IDUA and I2S, which are enzymes deficient in MPS I and II, respectively. This is a significant advantage for NBS operation in regions where MPS II is frequently identified, such as in Asian countries.

The scatter plot for each disease in this study indicated a tailed distribution. The CV values were mostly within 15% for the intraday assay, indicating that the assay provides stable values. In contrast, the interday fluctuations were relatively large. The average values of the control and the QC samples in each assay tended to behave in parallel. This fluctuation could be due to subtle changes in the methods used for the enzyme reaction, such as fluctuations in reaction temperature or reaction time from day to day. In the actual operation of NBS, quality control standards can be run each day along with newborn samples, and enzyme activities can be expressed as a percentage of the population mean or percentage of the daily population mean [31].

The mean enzyme activity for each MPS type in this study tended to be lower than those reported previously [18,29,31] (Table 2). Because the DBSs used in this study were frozen for more than 9 months after collection, prolonged storage may have led to some loss of enzyme activity. In this regard, the accumulation of data analyzing fresh DBSs is preferable, particularly for MPS II and IVA.

Comparison of enzyme activity in 23 patients with MPS with those of healthy subjects from NBS revealed that all patients displayed absent or low enzyme activity, which was nearly equivalent to that in blank DBS. Nevertheless, those derived from some patients with type II or type IVA overlapped with the lower-end values measured in control samples. This overlap between controls and patients can be explained by pseudo-deficiency (PD) in controls or sample deterioration resulting from prolonged storage. It has been reported that measuring enzyme activity by LC-MS/MS significantly reduces the number of PD cases compared to that using 4-methylumbelliferone [33]. However, it remains difficult to completely distinguish PD cases from controls [31]. It is also possible that the enzyme activity, which was originally in the lower limit of the normal range, decreased to the patient range because of long-term storage. The gradual decrease of lysosomal enzyme activity in DBS during long-term storage was reported previously [34,35]. Although the genetic analysis was not performed in the cases with overlapping values, no GAG accumulation was observed in the DBS of these samples [19], strongly suggesting that cases with low enzyme activities were not affected [7]. Indeed, clinical signs associated with MPSs have not been reported in these cases. Therefore, a combination of enzyme assays and GAG assays will provide precision screening with a minimum number of false-positive and false-negative data, as we described in our recent pilot study [7,10]. Further prospective pilot study of a large population is required to verify whether GAG assay and/or genetic analysis is better for the second-tier test in MPS screening.

A recent NBS study of 100,000 newborns using LC-MS/MS to assay the enzymes involved in MPS II, MPS IIIB, MPS IVA, MPS VI, and MPS VII revealed the following number of newborns with enzyme activities below the cutoffs: MPS II (*n* = 18), MPS IIIB (*n* = 0), MPS IVA (*n* = 8), MPS VI (*n* = 4), and MPS VII (*n* = 1) [31]. This is a manageable number of first-tier screen-positive samples. For MPS II and MPS VII, 2 of 18 screen-positives and 1 of 1 screen-positives, respectively, were likely to be disease-affected based on subsequent genotype analysis [31]. Zero of four MPS VI screen-positive samples were predicted to be true positives based on genotype, and no genotype data could be obtained for the MPS IVA screen-positives [31]. It is anticipated that second-tier analysis by measuring GAGs in DBSs will significantly reduce the false-positive rate. However, in the previous study, insufficient DBS material was available to test this idea.

The LC-MS/MS method described herein allowed for the simultaneous analysis of enzyme activity in MPSs. The frequency of each MPS and significance of each disease in NBS varies between countries. Another advantage of the LC-MS/MS method is that different diseases can be screened according to local, regional, or national needs.

## 4. Materials and Methods

### 4.1. Materials

DBS from 672 presumably healthy babies obtained by routine procedures were used as controls. After routine examination for NBS, the DBS was stored at −30 °C for approximately 9–12 months prior to analysis in the present study. DBS from 23 patients with MPS, all of whom were severe and had not been treated by either enzyme replacement therapy or hematopoietic stem cell transplantation, were analyzed. Patients included 2, 14, 1, 4, and 2 cases with MPS type I, II, IIIB, IVA, and VI, respectively. This study was approved by the Institutional Review Committee of Shimane University Faculty of Medicine (approval No.: 2475, approval date: 14 December 2016).

### 4.2. Reagents

The IS and substrate of each MPS were obtained from PerkinElmer (Waltham, MA, USA; Figure 3 and Table 3). Substrate concentrations were determined as described previously [30]. LC-MS grade methanol, acetonitrile, and ultrapure water were purchased from FUJIFILM Wako Pure Chemical Corporation (Osaka, Japan). DBS samples for quality control were provided by PerkinElmer. The reaction buffer was prepared by dissolving each substrate and IS corresponding to each type of MPS in 5 mM cerium acetate, 50 mM sodium acetate, and 0.042 mM D-saccharic acid 1-4-lactone at pH 5.0. The reaction buffer was stored at 4 °C and used within 2 weeks of preparation.

### 4.3. Sample Preparation Procedure

In each well of a 96-well assay plate, a single 3 mm DBS punch was added to 30 µL of the reaction solution. The plate was shaken at 250 rpm for 16 h at 37 °C. The plate was sealed with plate sealing film. The reaction was quenched by adding 100 µL of 50:50 methanol:ethyl acetate, and the reaction solution was transferred to a 96-well deep-well plate. Next, 400 µL of ethyl acetate and 200 µL of 0.5 M NaCl in water were added to the sample. After mixing by pipetting up and down 10 times, the plate was centrifuged for 5 min at 1000× *g* to separate the solvent layers. The upper layer (200 µL) was transferred to a new multi-well plate, which was dried under nitrogen flow at room temperature. Next, 100 µL water/acetonitrile (55/45) containing 0.1% formic acid was added and analyzed by LC-MS/MS.

### 4.4. LC-MS/MS Analysis

Samples were measured using a Nexera MP System utilizing the SIL-30ACMP Multi-Plate autosampler and LCMS-8050 triple quadrupole mass spectrometer (Shimadzu Corporation, Kyoto, Japan) equipped with an electrospray ionization-positive and -negative source. The LC column was a Phenomenex Kinetex XB-C18 150 × 2.1 mm, 1.7 µm (Phenomenex, Torrance, CA, USA), and the column oven temperature was 40 °C. Samples (10 µL) were injected and eluted with a binary solvent gradient of 0.1% formic acid in water (solvent A) and 0.1% formic acid in acetonitrile (solvent B) at a flow rate of 0.4 mL/min. Separation was conducted at 40 °C according to the following gradient program: 0–0.5 min, 30% (B) to 30% (B); 0.5–3.5 min, 30% (B) to 100% (B); 3.5–5.0 min, 100% (B); and 5.01–6 min, 100(B) to 30% (B). The desolvation line temperature was 250 °C, the heat block temperature was 100 °C, drying gas flow was 15.0 L/min, and nebulizer gas flow was 3.0 L/min. The precursor ion and product ion for each MPS type are shown in Table 4.

### 4.5. QC

Three separate QC samples, i.e., ‘Low’, ‘Middle’, and ‘High’, from PerkinElmer covered the range of enzyme activities that included the low levels observed in patients with MPS and the high levels detected in normal controls. Each QC sample was analyzed twice at the beginning and end of each batch.

### 4.6. Statistical Analysis

Significant differences were evaluated using SPSS (Statistical software, Chicago, IL, USA). Groups of patients with MPS II and IVA were compared with controls using t-tests. Results with *p* values of less than 0.05 were considered statistically significant.

## Figures and Tables

**Figure 1 ijms-21-02025-f001:**
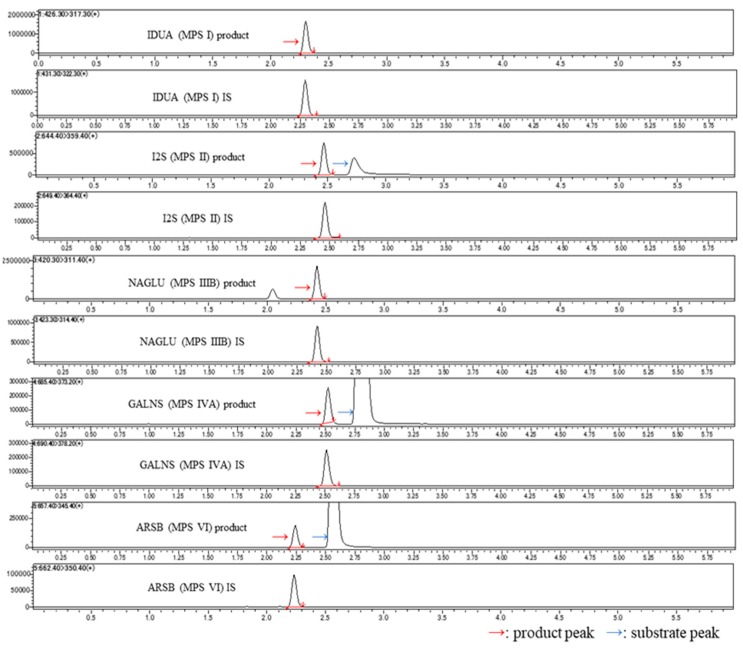
Chromatogram of substrate, reaction product, and internal standard (IS) for each mucopolysaccharidosis (MPS) enzyme. Red arrows represent the chromatogram for the product of enzyme reaction and blue arrows represent that for the substrate.

**Figure 2 ijms-21-02025-f002:**
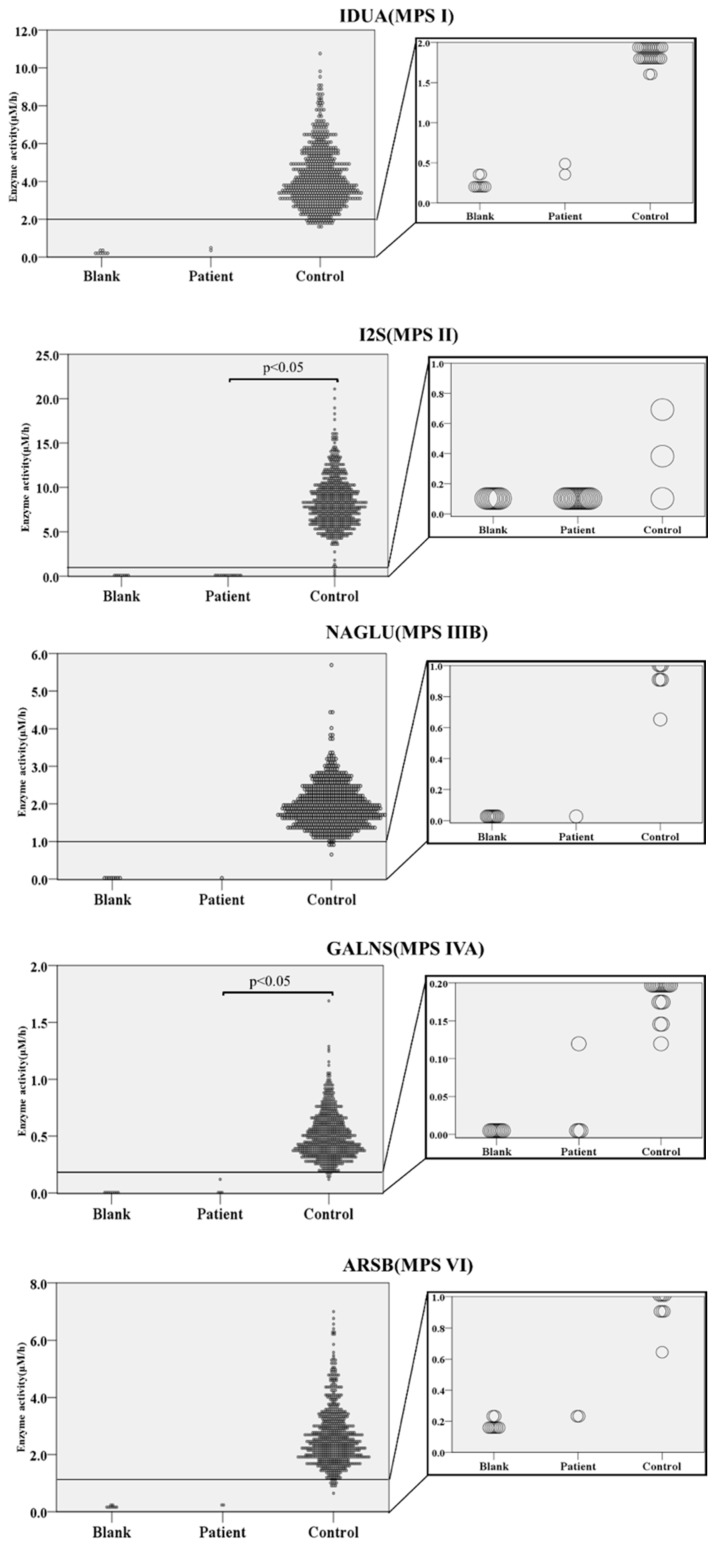
Violin plot of enzyme activities in dried blood spots (DBS). Blank; blank filter paper, patient(s); DBS from patient(s) with each type of MPS, controls; healthy subjects. Enlarged view of low-enzyme-activity area is shown by small window. Values of enzyme activities in blank (*n* = 5), in controls (*n* = 672), and patients with MPS: MPS I (*n* = 2), MPS II (*n* = 14), MPS IIIB (*n* = 1), MPS IVA (*n* = 4), and MPS VI (*n* = 2) were plotted. Mean enzyme activities of control samples were 4.19 µM/h in MPS I, 8.91 µM/h in MPS II, 1.96 µM/h in MPS IIIB, 0.50 µM/h in MPS IVA, 2.64 µM/h in MPS VI.

**Figure 3 ijms-21-02025-f003:**
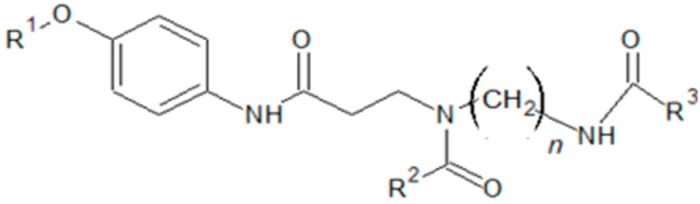
Common structure of each substrate and internal standard.

**Table 1 ijms-21-02025-t001:** Mean enzyme activity of QC and control (µM/h) and intraday and interday CV values (%).

		Day1	Day2	Day3	Day4	Day5	Interday CV
IDUA (MPS I)	Low	0.82 (15.98)	0.88 (2.43)	0.94 (3.04)	1.15 (2.49)	1.19 (13.62)	14.65
	Middle	5.02 (5.51)	5.12 (6.46)	6.13 (4.31)	7.13 (3.30)	7.04 (9.24)	14.78
	High	10.25 (5.40)	9.22 (5.23)	12.12 (5.85)	15.70 (11.19)	13.51 (4.84)	18.98
	Control	2.90	3.65	4.50	5.70	none	-
I2S (MPS II)	Low	0.58 (8.31)	0.54 (7.41)	0.68 (6.44)	0.75 (10.56)	0.66 (13.53)	11.82
	Middle	4.86 (7.55)	4.38 (2.73)	5.62 (2.84)	6.03 (5.55)	4.88 (7.30)	11.48
	High	10.45 (6.18)	10.13 (3.43)	11.86 (5.26)	12.53 (7.26)	8.74 (12.86)	12.44
	Control	6.63	7.13	9.05	10.76	none	-
NAGLU (MPS IIIB)	Low	0.24 (11.96)	0.24 (6.78)	0.23 (8.46)	0.25 (9.49)	0.28 (11.34)	6.85
	Middle	2.08 (6.19)	1.85 (10.41)	1.73 (7.76)	2.06 (10.84)	1.96 (7.37)	6.85
	High	4.85 (4.59)	4.22 (2.69)	4.29 (16.11)	4.84 (5.41)	4.77 (13.38)	6.06
	Control	1.68	1.98	1.93	2.25	none	-
GALNS (MPS IVA)	Low	0.10 (4.89)	0.18 (1.41)	0.19 (6.15)	0.22 (7.55)	0.19 (14.03)	23.13
	Middle	0.87 (12.55)	1.42 (7.66)	1.72 (8.74)	1.78 (9.72)	1.39 (4.67)	22.59
	High	1.88 (4.64)	3.27 (10.65)	3.86 (4.75)	4.04 (13.23)	2.78 (19.72)	24.73
	Control	0.45	0.42	0.51	0.60	none	-
ARSB (MPS VI)	Low	0.39 (9.40)	0.46 (9.17)	0.55 (6.20)	0.55 (14.17)	0.49 (11.55)	11.82
	Middle	2.13 (4.79)	2.39 (3.99)	3.20 (5.40)	3.35 (2.33)	2.90 (8.40)	16.66
	High	4.79 (7.09)	5.30 (9.60)	6.71 (2.64)	7.80 (12.14)	4.60 (24.74)	21.02
	Control	2.38	2.12	2.93	3.12	none	-

Mean enzyme activity in quality control (QC) and healthy subjects in each disease type analyzed on five different days. Enzyme activities of DBS in QC (*n* = 4) and healthy subjects (*n* = 164) on each day.

**Table 2 ijms-21-02025-t002:** Mean enzyme activity of healthy subjects (μM/h).

	IDUA	I2S	NAGLU	GALNS	ARSB
This study	4.19	8.39	1.96	0.5	2.64
Other studies	6.56 *	19.6 ** 16.52 ***	2.92 ***	2.1 ***	13.4 ** 14.6 ***

Enzyme activity in each disease type in the present study and previous reports. Reference: * [18] ** [29] *** [31].

**Table 3 ijms-21-02025-t003:** Chemical structure and final concentration of substrate and internal standard for each enzyme.

		Molecular Weight	Final Concentration (μM)	R^1^	R^2^	R^3^	*n*
MPS I (IDUA)	Substrate	601.65	249.82	α-Iduronosyl	Methyl	Phenyl	6
	Internal Standard	430.55	15.48	H	Methyl	d_5_-Phenyl	6
MPS II (IDS)	Substrate	767.23	470.01	α-Iduronosyl-2-sulfate	n-Butyl	Phenyl	6
	Internal Standard	648.34	5.14	α-Iduronosyl	n-Butyl	d_5_-Phenyl	6
MPS IIIB (NAGLU)	Substrate	622.76	501.19	α-*N*-Acetyl-glucosyl	n-Butyl	Ethyl	6
	Internal Standard	422.3	5.02	H	n-Butyl	d_3_-Ethyl	6
MPS IVA (GALNS)	Substrate	786.8	970.56	α-*N*-Acetyl-galactosyl-6-sulfate	n-Pentyl	Phenyl	6
	Internal Standard	689.39	4.84	α-*N*-Acetyl-galactosyl	n-Pentyl	d_5_-Phenyl	6
MPS VI (ARSB)	Substrate	689.39	4.84	α-*N*-Acetyl-galactosyl-4-sulfate	n-Pentyl	Phenyl	5
	InternalStandard	758.28	967.10	α-*N*-Acetyl-galactosyl	n-Butyl	d_5_-Phenyl	5

R^1^, R^2^, and R^3^ are the functional groups of each substrate and internal standard.

**Table 4 ijms-21-02025-t004:** Precursor ion and product ion of the reaction product and internal standard in each disease type.

		Precursor Ion (m/z)	Product Ion (m/z)
MPS I (IDUA)	Product	426.10	317.20
	Internal Standard	431.20	322.20
MPS II (IDS)	Product	644.32	359.23
	Internal Standard	649.35	364.26
MPS IIIB (NAGLU)	Product	420.20	311.20
	Internal Standard	423.20	314.20
MPS IVA (GALNS)	Product	685.38	373.25
	Internal Standard	690.41	378.28
MPS VI (ARSB)	Product	657.35	345.21
	Internal Standard	662.38	350.25

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
