# Peer review of "Validation of Liquid Chromatography-Tandem Mass Spectrometry-Based 5-Plex Assay for Mucopolysaccharidoses"

_ijms, 2020, doi:10.3390/ijms21062025_

Round 1

Author Response

Response to Reviewer 1 Comments

We appreciate the reviewer for taking the time to review our manuscript and for the constructive suggestions to improve our paper. The comments were seriously considered, and our point-by-point responses to each comment are described below. We think that those comments have allowed us to significantly improve our manuscript. We hope that we have adequately responded to the concerns raised by the reviewer.

In closing, let us thank you once again for your extremely cogent comments that have helped us improve the quality of our paper.

Point 1: Title, I recommend adding liquid chromatography into the title, since LC-MS/MS is used. 

Response 1: Thank you for your suggestion. We added liquid chromatography into the title. “Validation of liquid chromatography-tandem mass spectrometry-based 5-plex assay for mucopolysaccharidoses”

Point 2: Introduction, since several methods for MPSs have been reported, a brief summary/comparison on existing methods should be included. The authors should make it clear why they need to develop and validate the method in the manuscript. Is there any improvement on some aspects?

Response 2: Thank you for pointing out very important aspect. As the reviewer pointed out, the method in this study is an extension of Gelb's method. We have received information on the composition of the reaction cocktail from our co-researcher, Dr. Gelb.

We believe that our methods in this study differ in two points : 1) 5 MPS enzyme activities are analyzed in the same well of the plate, 2) in order to improve the quantification of enzyme activity, LC separation was improved compared to that of reported previously. As described in Lines 60-61, MPS I and MPS II enzyme activities were measured in separate assays in NBS. However, in Asian countries, including Japan, simultaneous measurement of enzyme activities for MPS I and II when performing MPS screening because MPS I and II are most frequent in these countries. This study demonstrated that simultaneous pre-processing and simultaneous analysis of MPS I and II are applicative, and that both MPS I and II patients can be identified by this method. In addition, the method reported by Liu et al. did not provide enough LC separation of the reaction products in our laboratory, which was required to measure the enzyme activities especially for MPS I and MPS III when analyzed. We believe that we have achieved more sensitive analysis by developing separation conditions. We added the above points in the revised manuscript(Lines 58-65).

Point 3: One important aspect of an enzyme activity assay is the substrate used. No such information is provided in the manuscript. The authors should provide this information for each substrate (name, structure, concentration) and the rationales behind the choice.

Response 3: Thank you for pointing this out. We agree with the reviewer. The description of the structure of the substrate was added as Table 3 (Lines 211).

Table 3. Chemical structure and final concentration of substrate and internal standard for each enzyme

Molecular weight

Final concentration (μM)

R1

Rï¼’

R3

n

MPS I
(IDUA)

Substrate 

601.65

249.82

α-Iduronosyl

Methyl

Phenyl

6

Internal Standard 

430.55

15.48

H

Methyl

d5-Phenyl

6

MPS II
(IDS)

Substrate 

767.23

470.01

α-Iduronosyl-2-sulfate

n-Butyl

Phenyl

6

Internal Standard 

648.34

5.14

α-Iduronosyl

n-Butyl

d5-Phenyl

6

MPS IIIB
(NAGLU)

Substrate 

622.76

501.19

α-N-Acetyl-glucosyl

n-Butyl

Ethyl

6

Internal Standard 

422.3

5.02

H

n-Butyl

d3-Ethyl

6

MPS IVA
(GALNS)

Substrate 

786.8

970.56

β-N-Acetyl-galactosyl-6-sulfate

n-Pentyl

Phenyl

6

Internal Standard 

689.39

4.84

β -N-Acetyl-galactosyl

n-Pentyl

d5-Phenyl

6

MPS VI
(ARSB)

Substrate 

689.39

4.84

β -N-Acetyl-galactosyl-4-sulfate

n-Pentyl

Pheny

5

Internal Standard 

758.28

967.10

β -N-Acetyl-galactosyl

n-Butyl

d5-Phenyl

5

The structural formula is a common structure. R1, R2, and R3 are the functional groups of each substrate and internal standard.

In accordance with the above changes, the Table 3 was changed to Table 4 as follows.

Table 4. Precursor ion and product ion of reaction product and internal standard in each disease type.

Precursor ion [m/z

Product ion [m/z

MPS I
(IDUA)

Product 

426.10

317.20

Internal Standard 

431.20

322.20

MPS II
(IDS)

Product 

644.32

359.23

Internal Standard 

649.35

364.26

MPS IIIB
(NAGLU)

Product 

420.20

311.20

Internal Standard 

423.20

314.20

MPS IVA
(GALNS)

Product 

685.38

373.25

Internal Standard 

690.41

378.28

MPS VI
(ARSB)

Product 

657.35

345.21

Internal Standard 

662.38

350.25

Point 4 Line 71-73, it is not clear to readers why chromatographic separation is essential here. I recommend the authors provide more details on “in source fragmentation” or cite a good reference for readers with no prior information. Similarly, in Line 124, the explanation is not clear.

Response 4: Thank you for your suggestion. We added the meaning of ISF in the Introduction in the manuscript. It was described that an importance of column separation to avoid the influence of ISF when doing MPS screening for MPS II, IV, and IV. The following references were cited (Lines 140).

Spacil, Z.; Tatipaka, H.; Barcenas, M.; Scott, C.R.; Turecek, F.; Gelb, M.H. High-Throughput Assay of 9 Lysosomal Enzymes for Newborn Screening. Clin. Chem. 2013, 59, 502–511.

Point 5: Line 120-121, “This work is an extension of a previous study by Gelb et al. using LCMS/MS to simultaneously analyze the same MPSs in a multiplex assay [31]”. I don’t agree with this statement. The method in ref 31 analyze more enzyme than the method in this manuscript and the LC-MS/MS is much shorter (2 min vs 6 min). I don’t see any improvement in the authors’ method.

Response 5: It is an important point. The advantages of method in present study are that enzyme activities for 5 MPSs including MPS I and II are pre-treated in the same well and analyzed simultaneously, and that the developed analytical method improved sensitivity by changing the analytical time from 2 min to 6 min. As mentioned above, I have specified that in Introduction (Lines 58-62, Lines 66-70). And the description about shape and separation of chromatogram obtained have been described in Results (Lines 80-81).

Point 6: Line 130-132, I don’t think mentioning a method with any details/data is valuable. 

Response 6: We agree with the reviewer. The sentences that mention about the short-time method were deleted.

Point 7: Line 136-138, “This fluctuation could be due to subtle changes in the methods used for the enzyme reactions, such as fluctuations in reaction temperature or reaction time from day to day.”  From the methods part, the reaction temperature and time are 37 °C and 16 h. Where do the fluctuations come from?

Response 7: Thank you for your question. We are considering an influence by environmental temperature, sample temperature, etc. The detailed cause of these phenomena is not well understood in this study, but it seems to have a similar tendency in other studies.

Reviewer 2 Report

The expanding number of treatments available for MPS is placing an unprecedented demand for accurate, timely (and, ideally, cost-effective) diagnosis of these diseases, as it is well-recognized that the best outcomes are seen with early initiation of appropriate therapies. Thus, any novel method, which contributes to this overall need, is a valuable contribution to the LSD field.

Here, Oguni and co-workers, present a 5-plex LC-MS/MS method for the simultaneous measurement of the activities of 5 MPS-related enzymes: IDUA, I2S, NAGLU, GALNS and ARSB.

The method itself has its value. Yet, I do have some concerns on the paper:

The method the authors describe is basically “an extension” (as the authors themselves point out) of that of Gelb et al.. Yet, the reference they quote under brackets every time they elaborate on the subject refers to a paper with a 7-plex approach for MPSs + CLN (Liu et al., 2017; listed in the manuscript as reference 31).  Is this the right reference? If so, the authors should not refer to it as Gelb et al. but Liu et al. I understand they mean Michael Gelb’s lab, WA. Yet, for the regular reader it can get a bit confusing. Furthermore, there’s a recently published paper by Scott and co-workers on a 5-plex approach for MPSs (also from Michael Gelb’s lab), which describes a method for the simultaneous screening of the exact same MPS reported in the manuscript, which is here under review. So, I would strongly recommend the authors to use the first author name whenever quoting papers on the methods to avoid confusion (page 2, line 54-56 and page 5, lines 120-121). Plus, I feel they should make it clear right from the beginning the method they are relying on: is it the same used by Liu et al., 2017? Is it that of Scott et al., 2020? Or is it an in-house developed assay based on either of the previous ones? This should be clearly disclosed; otherwise, the readers are forced to go check the original references and compare the methods point-to-point.

Concerning the overall results, I would appreciate if the authors further elaborated on the small numbers of patient samples for some of the assessed MPSs (MPS I, n=2; MPS IIIB, n=1 and MPS VI, n=2). Do they think this small numbers may influence the overall conclusions? How confident can they be on a ‘typical’ enzyme activity value for a given disease if only one patient has been analysed? This should be discussed.

Also, I believe the fact that there’s some overlaps between Ct and patient values for MPS II and IVA would be worth extra discussion. Most importantly, I think it would worth an extra effort to check whether those values could be explained by pseudodeficiencies. It does seem a rationale explanation. And yet, it wouldn’t be hard to test it experimentally and I think the paper would largely benefit from that extra study: recalling the samples with overlapping values for a sort of second-tier test. They could either use DBS to extract gDNA and assess their genotypes to check whether that would be the case or run an assay on accumulated GAG.

Another issue that deserves further discussion is the effect of prolonged storage of DBS and its effect on enzyme activity. Isn’t it possible that this is one of the major issues underlying the almost complete absence of enzyme activity? Can’t this be an underestimate of the enzyme activity interval for patients? This should be discussed.

Minor issues:

* While introducing the diseases and their related enzymes, I think the authors should add the OMIM# reference number for each MPS disease, as well as the E.C. numbers that identify each enzyme.

* On page 3, Figure 2, the authors should change the sequence of the first two violin plots, starting with IDUA (MPS I) and presenting the I2S (MPS II) subsequently.

* The authors should disclose (or further elaborate) on the sentence that opens the paper’s last paragraph (apart from the M&M section): “The LC-MS/MS method described herein allowed for simultaneous analysis of enzyme activity in other LSDs as well as in MPSs.” Why do they refer to ‘other LSDs’? That has not been discussed in the manuscript.

Author Response

Response to Reviewer 2 Comments

We appreciate the reviewer for taking the time to review our manuscript and for the constructive suggestions to improve our paper. The comments were seriously considered, and our point-by-point responses to each comment are described below. We think that those comments have allowed us to significantly improve our manuscript. We hope that we have adequately responded to the concerns raised by the reviewer.

In closing, let us thank you once again for your extremely cogent comments that have helped us improve the quality of our paper.

Point 1: The method the authors describe is basically “an extension” (as the authors themselves point out) of that of Gelb et al.. Yet, the reference they quote under brackets every time they elaborate on the subject refers to a paper with a 7-plex approach for MPSs + CLN (Liu et al., 2017; listed in the manuscript as reference 31).  Is this the right reference? If so, the authors should not refer to it as Gelb et al. but Liu et al. I understand they mean Michael Gelb’s lab, WA. Yet, for the regular reader it can get a bit confusing. Furthermore, there’s a recently published paper by Scott and co-workers on a 5-plex approach for MPSs (also from Michael Gelb’s lab), which describes a method for the simultaneous screening of the exact same MPS reported in the manuscript, which is here under review. So, I would strongly recommend the authors to use the first author name whenever quoting papers on the methods to avoid confusion (page 2, line 54-56 and page 5, lines 120-121). Plus, I feel they should make it clear right from the beginning the method they are relying on: is it the same used by Liu et al., 2017? Is it that of Scott et al., 2020? Or is it an in-house developed assay based on either of the previous ones? This should be clearly disclosed; otherwise, the readers are forced to go check the original references and compare the methods point-to-point.

Response 1: Thank you for pointing out an important issue. The composition of the cocktail used in the present study was the same as that reported by Liu et al., but the analytical method was different. We aimed to achieve accurate analysis by clear LC-separation of each reaction product. Therefore, our method took the analysis time for six minutes. We added this difference in the text.

Point 2: Concerning the overall results, I would appreciate if the authors further elaborated on the small numbers of patient samples for some of the assessed MPSs (MPS I, n=2; MPS IIIB, n=1 and MPS VI, n=2). Do they think this small numbers may influence the overall conclusions? How confident can they be on a ‘typical’ enzyme activity value for a given disease if only one patient has been analyzed? This should be discussed.

Response 2: All MPS patient specimens analyzed were untreated and severe. As the reviewer pointed out, this study had only two samples of MPS I and MPS IV and one sample of MPS IIIB. We agree that it is difficult to generalize from these results. However, we thought that it was a certain meaning that our method was able to identify the positive patient. Of course, we believe that further verification is necessary. We mentioned it in the manuscript (Lines 177-179).

Point 3: Also, I believe the fact that there’s some overlaps between Ct and patient values for MPS II and IVA would be worth extra discussion. Most importantly, I think it would worth an extra effort to check whether those values could be explained by pseudodeficiencies. It does seem a rationale explanation. And yet, it wouldn’t be hard to test it experimentally and I think the paper would largely benefit from that extra study: recalling the samples with overlapping values for a sort of second-tier test. They could either use DBS to extract gDNA and assess their genotypes to check whether that would be the case or run an assay on accumulated GAG.

Response 3: We agree with the opinion of the reviewer. As the reviewer pointed out, the overlap between patients and control groups in enzyme activities for MPS II and MPS IVA should be discussed sufficiently. We wanted to perform both genetic analysis and GAG analysis to confirm whether the overlap case was PD or not, but we did not have the consent to use the DBS for genetic analysis. Therefore, we analyzed GAG in DBS and confirmed that there was no GAG accumulation in these DBS samples from controls (Line158-160). The need for future research to proactively study both GAG and genetic analysis was also mentioned in Discussion (Lines 177-179).

Point 4 Another issue that deserves further discussion is the effect of prolonged storage of DBS and its effect on enzyme activity. Isn’t it possible that this is one of the major issues underlying the almost complete absence of enzyme activity? Can’t this be an underestimate of the enzyme activity interval for patients? This should be discussed.

Response 4: We agree with the reviewer. We mentioned about the effect of prolonged storage of DBS in measuring enzyme activity in the manuscript (Lines 170-171).

Minor issues:

Point 5: While introducing the diseases and their related enzymes, I think the authors should add the OMIM# reference number for each MPS disease, as well as the E.C. numbers that identify each enzyme.

Response 5: OMIM numbers for each MPS type and EC numbers for each enzyme were added in the manuscript.

Point 6: On page 3, Figure 2, the authors should change the sequence of the first two violin plots, starting with IDUA (MPS I) and presenting the I2S (MPS II) subsequently.

Response 6: Thank you for pointing out our mistakes. We changed Fig. 2 in the correct order.

Point 7: The authors should disclose (or further elaborate) on the sentence that opens the paper’s last paragraph (apart from the M&M section): “The LC-MS/MS method described herein allowed for simultaneous analysis of enzyme activity in other LSDs as well as in MPSs.” Why do they refer to ‘other LSDs’? That has not been discussed in the manuscript.

Response 7: Thank you for your comment. We deleted that sentence.

Reviewer 3 Report

The Mucopolysaccharidoses represent a group of very severe inherited lysosomal storage disorders with involvement of multiple organs. Life expectancy of patients affected from these diseases are very short, so an early and rapid diagnosis are fundamental. In the manuscript, the authors described an improved newborn screening approach of Mucopolysaccharidoses based on a simultaneous enzymatic activity assessment of five lysosomal enzymes using LC-MS/MS.

In particular, the authors showed the advantages of this new NBS methodology in terms of enzymatic values stability and reduction of time-consuming respect to the assessment of single enzymatic activities. This technique might allow an improvement of the newborn screening of MPSs especially in countries in which it is registered a high frequency of these diseases.

Although the authors explained the different advantages of this new methodology for the evaluation of lysosomal enzymatic activity, some suggestions might be considered in order to ameliorate the manuscript.

Minor revisions:

-In order to improve the statistical significance of results, the author should increase the number of blood samples from the following MPS patients: MPS-I, MPS-IIIB and MPS-VI. It should be better to have at least 3-4 samples for each MPS disease tested.

- In the discussion the authors hypothesized that the deterioration of blood samples of some MPS patients caused an overlapping of their enzymatic values ​​with those of control samples analyzed. In order to verify the impact of storage condition on the success of enzymatic activity assessment and to confirm the efficacy of this technique, the authors should measure new fresh blood samples from MPS-II and MPS-IVA patients.

Author Response

Response to Reviewer 3 Comments

We appreciate the reviewer for taking the time to review our manuscript and for the constructive suggestions to improve our paper. The comments were seriously considered, and our point-by-point responses to each comment are described below. We think that those comments have allowed us to significantly improve our manuscript. We hope that we have adequately responded to the concerns raised by the reviewer.

In closing, let us thank you once again for your extremely cogent comments that have helped us improve the quality of our paper.

Point 1: In order to improve the statistical significance of results, the author should increase the number of blood samples from the following MPS patients: MPS-I, MPS-IIIB and MPS-VI. It should be better to have at least 3-4 samples for each MPS disease tested.

Response 1: Thank you for pointing out an important issue. As the reviewer pointed out, the number of samples were small to evaluate statistical significance, but there were only 1-2 samples of MPS I, MPS IIIB and MPS VI in this study. We believe that further verification is necessary, so we mentioned it in the manuscript (Lines 177-179).

Point 2: In the discussion the authors hypothesized that the deterioration of blood samples of some MPS patients caused an overlapping of their enzymatic values with those of control samples analyzed. In order to verify the impact of storage condition on the success of enzymatic activity assessment and to confirm the efficacy of this technique, the authors should measure new fresh blood samples from MPS-II and MPS-IVA patients.

Response 2: Thank you for your important point. We mentioned about the effect of prolonged storage of DBS in measuring enzyme activity in the manuscript (Lines 170-171). Also, as the reviewer pointed out, we believe that the prospective pilot study is necessary, so we mentioned in the manuscript (Lines 176-178).

Round 2

Reviewer 1 Report

Comments to the Author:

The authors responded nicely to all my comments.

Minor comments:

  1. Line 28-29, units for the enzyme activities are missing for some of the values
  2. Line 60-62, the explanation of in-source fragmentation is a little confusing.
  3. Line 81, spectrums => chromatograms
  4. Line 172, thr => the

Author Response

Response to Reviewer 1 Comments

We appreciate the reviewer for taking the time to review our manuscript again. Our point-by-point responses to each comment are described below. We hope that we have adequately responded to the concerns raised by the reviewer.

In closing, let us thank you once again for your extremely cogent comments that have helped us improve the quality of our paper.

Point 1: Line 28-29, units for the enzyme activities are missing for some of the values

Response 1: Thank you for pointing out our mistakes. We added units to all values.

Point 2: Line 60-62, the explanation of in-source fragmentation is a little confusing.

Response 2: Thank you for your suggestion. We rewrote the explanation of in-source fragmentation to make it clearer.

Point 3: Line 81, spectrums => chromatograms

Response 3: Thank you for pointing this out. We corrected.

Point 4 Line 172, thr => the

Response 4: Thank you for pointing this out. We corrected the typos.

Reviewer 2 Report

The authors have either addressed the points raised in the previous round of review or provided a solid explanation of doubts.

I believe the current version is signifficantly improved. 

Author Response

Response to Reviewer 2 Comments

We appreciate the reviewer for taking the time to review our manuscript again.

We are very pleased to receive the feedback that our response was adequately responded to the reviewer. We believe that your comments have allowed us to significantly improve our manuscript.

We carefully proofread the revised manuscript again to make sure all typos are corrected, which are highlighted in blue in the text.

In closing, let us thank you once again for your extremely cogent comments that have helped us improve the quality of our paper.